# Impact of Levetiracetam Treatment on 5-Aminolevulinic Acid Fluorescence Expression in IDH1 Wild-Type Glioblastoma

**DOI:** 10.3390/cancers14092134

**Published:** 2022-04-25

**Authors:** Johannes Wach, Ági Güresir, Motaz Hamed, Hartmut Vatter, Ulrich Herrlinger, Erdem Güresir

**Affiliations:** 1Department of Neurosurgery, University Hospital Bonn, 53127 Bonn, Germany; agi.gueresir@ukbonn.de (Á.G.); motaz.hamed@ukbonn.de (M.H.); h1artmut.vatter@ukbonn.de (H.V.); erdem.gueresir@ukbonn.de (E.G.); 2Division of Clinical Neurooncology, Department of Neurology and Centre of Integrated Oncology, University Hospital Bonn, 53127 Bonn, Germany; ulrich.herrlinger@ukbonn.de

**Keywords:** glioblastoma, 5-ALA, antiepileptic drugs, levetiracetam

## Abstract

**Simple Summary:**

The amino acid 5-aminolevulinic acid (5-ALA) is the benchmark regarding intraoperative imaging tools for glioblastoma (GB) surgery, and is known to facilitate the extent of resection, which results in an enhanced 6 month progression-free survival rate. Recent in vitro studies suggest that antiepileptic drugs (AEDs) result in a reduction in the fluorescence quality in gliomas. To date, there is no large clinical series investigating this issue in a homogeneous cohort. Approximately 25% of all GB patients have a symptomatic epilepsy as the initial symptom at presentation. Hence, this potential dilemma is of paramount importance. We found that the preoperative intake of levetiracetam is a significant risk factor for reduced intraoperative fluorescence in IDH1 wild-type GBs. We believe that this issue must be considered in future external validations, and physicians must carefully evaluate the indication of levetiracetam and avoid a prophylactic levetiracetam treatment in terms of the suspected diagnosis of glioblastoma.

**Abstract:**

The amino acid 5-aminolevulinic acid (5-ALA) is the most established neurosurgical fluorescent dye and facilitates the achievement of gross total resection. In vitro studies raised concerns that antiepileptic drugs (AED) reduce the quality of fluorescence. Between 2013 and 2018, 175 IDH1 wild-type glioblastoma (GB) patients underwent 5-ALA guided surgery. Patients’ data were retrospectively reviewed regarding demographics, comorbidities, medications, tumor morphology, neuropathological characteristics, and their association with intraoperative 5-ALA fluorescence. The fluorescence of 5-ALA was graded in a three point scaling system (grade 0 = no; grade 1 = weak; grade 2 = strong). Univariable analysis shows that the intake of dexamethasone or levetiracetam, and larger preoperative tumor area significantly reduce the intraoperative fluorescence activity (fluorescence grade: 0 + 1). Multivariable binary logistic regression analysis demonstrates the preoperative intake of levetiracetam (adjusted odds ratio: 12.05, 95% confidence interval: 3.91–37.16, *p* = 0.001) as the only independent and significant risk factor for reduced fluorescence quality. Preoperative levetiracetam intake significantly reduced intraoperative fluorescence. The indication for levetiracetam in suspected GB should be carefully reviewed and prophylactic treatment avoided for this tumor entity. Future comparative trials of neurosurgical fluorescent dyes need a special focus on the influence of levetiracetam on fluorescence intensity. Further trials must validate our findings.

## 1. Introduction

Gliomas represent the most common intracranial neoplasms and account for 70% of all primary brain tumors [1,2]. IDH1 wild-type glioblastoma (GB) is the most common malignant brain tumor, and the fifth edition of the World Health Organization (WHO) classification system classifies IDH1 wild-type gliomas as WHO grade 4 [3].

Benchmark treatments include a maximum complete resection, with the preservation of neurological functioning, and adjuvant radiochemotherapy [4]. Despite emerging data demonstrating the prognostic benefits of evidence-based concomitant chemoradiotherapy regimens, including lomustine-temozolomide (TMZ) and temozolomide, phase III trials found that the median overall survival (OS) in GB patients with a hypermethylated O-6-methylguaninen-DNA methyltransferase (MGMT) promotor, and treated with standard TMZ-based radiochemotherapy, is 23.4–31.4 months [5,6,7]. Further established prognostic factors for long-term overall survival in GB are age at diagnosis, baseline Karnofsky Performance Status (KPS), and the extent of resection (EoR) [8,9,10,11].

For patients with newly diagnosed GB, maximum EoR equates to a significant enhancement in overall survival. Hence, extent of resection thresholds between 95% and 98% are frequently recommended [11,12]. However, maximum cytoreductive surgery, while preserving a good neurologic outcome, is a major task in neuro-oncological surgery. Intraoperative imaging methods, such as intraoperative magnetic resonance imaging (iMRI) [13,14,15] and 5-aminolevulinic acid (5-ALA), are powerful tools used to achieve high rates of gross total resection (GTR) in glioblastoma surgery [16].

A 5-ALA-guided surgery induces protoporphyrin IX (PpIX) fluorescence, which results in a significantly higher extent of resection and prolonged progression-free survival in GB [16,17,18,19,20,21]. The precise mechanism resulting in the accumulation of PpIX in GB cells, so far, remain unclear. A multifactorial impact was suggested to influence the intraoperative fluorescence quality. For instance, increased metabolism, the up-regulation of porphyrin-producing enzymes, reduced metabolism of iron within glioma cells, and a reduction in activity of the enzyme ferrochelatase, which converts the visible and fluorescing PpIX into heme, are suggested as key mechanisms of 5-ALA application in glioblastoma surgery [22,23,24]. The exogenously administered 5-ALA is absorbed into the cytoplasm of the mitochondria and used as a substrate for PpIX [25,26]. PpIX acts as a protein-bound prosthetic group in mitochondrial respiratory chain complexes. In vitro studies reveal that antiepileptic drugs (AEDs), including levetiracetam, might injure the mitochondrial membrane, which results in an inhibition of PpIX synthesis in glioblastoma cells [27,28]. Hence, there is potential conflict between AED treatment and 5-ALA application in the surgical treatment of GB.

Seizures are frequently observed as the first symptom of GB, and the literature reports that between 25% and 50% of the patients suffer from symptomatic epilepsy at presentation [29]. For this type of seizure, the International League Against Epilepsy (ILAE) recommends levetiracetam as a class A efficacy AED [30]. Hence, the potential interaction between levetiracetam, as the most common prescribed AED in this condition, and the application of 5-ALA, concerns a large group of GB patients and might leave physicians on the horns of a dilemma.

The present investigation aimed to analyze the influence of AEDs, and in particular levetiracetam, on the presence and quality of visible 5-ALA fluorescence in surgery for IDH1 wild-type glioblastoma.

## 2. Materials and Methods

### 2.1. Study Design and Patient Characteristics

Between May 2013 and December 2018, 381 GB patients underwent surgical therapy for high-grade gliomas, classed as WHO grade 4, in the institutional neurosurgical center. A review of patient data was retrospectively performed, after institutional review board approval was obtained. The criteria for inclusion in this study were histopathologically confirmed IDH1 wild-type glioblastoma, primary diagnosis, an age greater than 18 years, 5-ALA application, description of fluorescence quality, and treatment via a neurosurgical resection. The following patients (*n* = 206) were excluded: patients who underwent biopsy without the application of 5-ALA; if an MRI showed multiple or bilateral disease; or if the physical status was graded as KPS < 60% [31]. Recurrent malignant glioma patients were excluded, due to potential false positive fluorescence after adjuvant therapies [32,33]. In 40 cases, 5-ALA was not applied, due to initially suggested brain metastases (*n* = 33), no visible Gd enhancement in preoperative MRI (*n* = 6), and pregnancy (*n* = 1). IDH1 mutated GBs were also excluded, to be in line with the fifth edition of the WHO classification system, and due to observed differences regarding fluorescence quality in IDH1 wild-type and mutated GBs [34]. Figure 1 summarizes the selection process of the present study cohort.

### 2.2. Surgical Procedure

Routine navigation head Gadolinium (Gd)-enhanced MRIs were routinely performed within 48 h before surgery. The 5-Aminolevulinic acid was administered orally as a single dose (20 mg/kg body weight, Gliolan; Medac GmbH, Wedel, Hamburg, Germany), on average 3 h preoperatively. Neurosurgical white-light resection was performed under neuronavigation guidance (BrainLAB Curve, BrainLAB AG, Feldkirchen, Bavaria, Germany). During surgery, the PENTERO 800 microscope (Zeiss, Oberkochen, Germany) was routinely switched to violet–blue excitation light, to visualize 5-ALA fluorescence positive areas in a dark operative room, which is necessary to reduce biases from external sources of light. The neurosurgical microscope included a fluorescent 400 nm UV light module and specified filters to enable 5-ALA visualization. When the neurosurgeon assumed that gross total resection of the tumor was achieved, hemostasis was performed. Afterward, the resection cavity was again examined using 5-ALA, and suspicious areas of residual tumor tissue were resected. Every single procedure was recorded and reviewed by a senior neurosurgeon. Fluorescence quality was visually evaluated by senior neurosurgeons, and classified according to the established 3-stage scale system of Stummer et al. [35]. The following classification was used: grade 0 constituted no fluorescence; grade 1 constituted weak fluorescence; and grade 2 constituted strong fluorescence. Figure 2 displays the applied classification system of 5-ALA fluorescence quality measurement scale, according to Stummer et al. [35]. Postoperative MRIs were performed within 72 h of resection, by a senior neuroradiologist, to evaluate the extent of resection [31]. Gross total resection was defined as a resection without residual Gd enhancement, whereas subtotal resection was considered as any resection with residual Gd enhancement and an extent of resection ≥ 90% [36].

### 2.3. Histopathology

Histopathological grading was performed based on the 2016 WHO criteria [37]. IDH1 mutated GBs were not included in this analysis. Immunohistochemical investigation for mutant IDH1 (R132H) was performed. In patients < 55 years, and with negative immunohistochemistry, further mutational testing was performed. Hence, all WHO grade 4 tumors are in keeping with the requirements of the fifth edition of the WHO classification system [3]. Paraffin sections were stained with hematoxylin and eosin (H & E). Tumor specimens were immunohistochemically examined using the molecular immunology borstel-1 (MIB-1) antibody, glial fibrillary acidic protein (GFAP), and IDH1 [31]. MGMT promoter status was investigated and reported according to Hegi et al. [38]. The O-6-methylguanine-DNA promoter methylation was routinely investigated using pyrosequencing, as described previously [39].

### 2.4. Clinical Data Recording and Analysis

The following preoperative patient characteristics were recorded: age, sex, Karnofsky Performance Status, American Society of Anesthesiologists classification (ASA), body mass index (BMI), medical comorbidities, and medication. Since the primary objective of the present investigation was to analyze the impact of AEDs on 5-ALA fluorescence quality in GB surgery, each type of AEDs administered in each patient was documented.

Tumor characteristics were investigated based on a measurement of the tumor area, which was calculated in mm^2,^ based on the two largest tumor diameters perpendicular to each other on the axial Gd-enhanced, T1-weighted MR images in the preoperative imaging examination [31,40]. Perilesional edema was measured as the maximum extent of the hyperintense T2 signal intensity on the tumor margin in the preoperative MRI [31,41].

### 2.5. Statistical Analysis

Data were organized and analyzed using SPSS for Windows (version 27.0; IBM Corp, Armonk, New York, NY, USA). Receiver operating characteristic (ROC) curves were constructed for tumor areas in the prediction of fluorescence quality. Cut-off values for the variables age and MIB-1 labeling index were set at 65 years and 20%, based on previous studies [42,43,44]. Normally distributed data were reported as mean with the standard deviation (SD). Preoperative demographic data, medications, tumor features, and neuropathological data were compared between fluorescence quality grades using Fisher’s exact test (two-sided) for categorical data, and analysis of variance (ANOVA) and independent *t*-test for continuous data. For a poor fluorescence quality (grade 0 + 1), multivariable binary logistic regression analysis of predictors was performed. A *p*-value < 0.05 was defined as statistically significant. MGMT and MIB-1 index were included in the multivariable analysis, due to their described relationship with age and 5-ALA fluorescence [45,46]. Further subgroup analyses (using uni- and multivariable analyses) of fluorescence quality in patients treated with either levetiracetam or no AEDs were performed.

## 3. Results

### 3.1. Patient Characteristics

A total of 175 patients underwent 5-ALA-guided surgery for IDH1 wild-type GB at our department between May 2013 and December 2018. The median age is 66 years (IQR 56*–*73), and the study includes 66 females (37.7%) and 109 males (62.3%; male/female ratio 1.65:1). The median preoperative Karnofsky performance scale (KPS) at presentation is 90 (IQR 80*–*90). A total of 52 patients (52/175; 29.7%) suffer from symptomatic epilepsy at presentation. The most common seizure type is generalized epilepsy (24/175; 13.7%). In 49 out of 52 patients with a baseline epilepsy, AED treatment is already introduced prior to surgery. Levetiracetam (43/175; 24.6%) is the most frequently prescribed AED. Strong 5-ALA fluorescence is observed in 142 (81.1%) patients, whereas in 33 (18.9%) patients either no, or only weak, fluorescence quality is found. Further characteristics are summarized in Table 1.

### 3.2. Clinical, Imaging, and Neuropathological Characteristics among Fluorescence Grades

Fluorescence grades 0, 1, and 2 are observed in 16 (9.1%), 17 (9.7%), and 142 (81.1%) patients, respectively. Patients with a stronger fluorescence quality tend to be older, compared to fluorescence grade 1 and 0 patients (*p* = 0.06). The intake of AED and dexamethasone prior to surgery is significantly more often observed among the patients in whom either no, or only weak, intraoperative fluorescence quality is observed. A total of 26 out of 33 patients with either no, or only weak, fluorescence grades (grades 0 + 1) take antiepileptic drugs prior to surgery, whereas only 7 patients with fluorescence grades 0 or 1 have no AEDs in their preoperative medication (*p* = 0.001*).* Strong intraoperative fluorescence grade 2 is also significantly more often observed in patients who take no dexamethasone preoperatively. The intake of dexamethasone is not significantly associated with the intake of AEDs. A total of 19 patients (19/52; 36.5%) in the AED group simultaneously take dexamethasone, whereas 56 patients (56/123; 45.5%) without the intake of AEDs are also under prescription of dexamethasone preoperatively (Fisher’s exact test (two-sided): *p* = 0.32). Furthermore, the tumor area is significantly larger in patients with fluorescence grade 2, in response to UV light excitation (mean ± SD in fluorescence grades 2, 1, and 0: 1553.7 ± 1047.8 vs. 1197.6 ± 921.4 vs. 834.1 ± 533.8; *p* = 0.03). MGMT promoter methylation status and proliferative potential, according to the MIB-1 labeling index, are homogeneously distributed among the fluorescence grades. Table 2 summarizes the characteristics and results among the fluorescence quality grades.

### 3.3. Association between Antiepileptic Drug Treatment and Intraoperative 5-ALA Fluorescence

Univariable analysis reveals that the intake of antiepileptic drugs, or dexamethasone, and the preoperative tumor area is significantly associated with the intraoperative 5-ALA fluorescence quality. The ROC curve (Appendix A) is constructed and the AUC of the tumor area in the prediction of a poor fluorescence grade (0 + 1) is determined. The AUC for tumor area is 0.66 (95% CI: 0.55*–*0.76, p = 0.009) The sensitivity and specificity of a baseline tumor area, with an optimum cut-off set at ≤967.4 mm^2^, for predicting a poor fluorescence quality is 61.0% and 66.2%, respectively (Youden’s index: 0.27). Multivariable binary logistic regression analysis is conducted to determine independent risk factors for poor intraoperative 5-ALA fluorescence (fluorescence grades 0 + 1). Multivariable analysis is performed with consideration of the MIB-1 index (<20/≥20%), MGMT promoter status (hypermethylated/non-hypermethylated), preoperative intake of dexamethasone (yes/no), preoperative tumor area (≤967.4/>967.4), age (<65/≥65), and preoperative intake of antiepileptic drugs (yes/no). The multivariable analysis reveals that the intake of antiepileptic drugs (adjusted odds ratio: 14.7, 95% confidence interval: 4.65*–*46.58, *p* = 0.001) is an independent predictor of a poor intraoperative 5-ALA fluorescence (fluorescence grades: 0 + 1) in surgery for GB. Figure 3 displays the results of the multivariable analysis.

### 3.4. Specific Impact of Levetiracetam on Intraoperative 5-ALA Fluorescence

A total of 43 (87.76%) patients of the 49 patients in the AED group take levetiracetam. Against this backdrop, the previous findings regarding the influence of AED on fluorescence quality are predominantly based on the influence of levetiracetam. Therefore, a specific subgroup analysis of patients who either take levetiracetam, or no AEDs, is performed. A total of 169 patients are included in this analysis. Fluorescence grades 0, 1, and 2 are observed in 12 (7.1%), 16 (9.5%), and 141 (83.4%) patients, respectively. Patients with stronger ALA fluorescence quality (grade 2) are significantly older compared to fluorescence 0 patients (*p* = 0.002). The intake of levetiracetam and dexamethasone prior to surgery is significantly more often observed among the patients in whom either no, or only weak, intraoperative fluorescence quality is observed. A total of 22 (78.6%) out of 28 patients with either no, or only weak, fluorescence grades (grades 0 + 1) take levetiracetam prior to surgery, whereas only 6 patients (21.4%) with fluorescence grades 0 or 1 have no levetiracetam prescription in their preoperative medication (*p* = 0.001*).* Strong intraoperative fluorescence grade 2 is also significantly more often observed in patients who do not take dexamethasone preoperatively. Moreover, the tumor area is significantly larger in patients with fluorescence grade 2 in response to UV light excitation (mean ± SD in fluorescence grades 2 vs. 0: 1550.9 ± 1048.5 vs. 830.7 ± 531.8; independent *t*-test: *p* = 0.002). The MGMT promoter methylation status and proliferative potential, according to the MIB-1 labeling index, are homogeneously distributed among the fluorescence grades. Table 3 summarizes the characteristics and results among the fluorescence quality grades.

Univariable analysis reveals that the intake of levetiracetam, intake of dexamethasone, age, and the preoperative tumor area are significantly associated with the intraoperative 5-ALA fluorescence quality. Hence, further multivariable analysis of intraoperative fluorescence quality is also performed for patients treated either with levetiracetam or no AEDs. Multivariable analysis, with consideration of levetiracetam intake, age at diagnosis, tumor area, intake of dexamethasone, MGMT promoter status, and MIB-1 labeling index, is performed. The MIB-1 labeling index and MGMT promoter status are included because of their previously described association with intraoperative fluorescence quality. The multivariable analysis shows that the intake of levetiracetam (adjusted odds ratio: 12.05, 95 % confidence interval: 3.91–37.16, *p* = 0.001) is an independent predictor of a poor intraoperative 5-ALA fluorescence (fluorescence grades: 0 + 1) in surgery for GB. Figure 4 shows the results of the multivariable analysis.

### 3.5. Impact of Intraoperative Fluorescence Quality on Extent of Resection and Influence of AEDs, and Levetiracetam on Overall Survival

Patients with a poor fluorescence quality (grades 0 & 1) have a GTR in 19 cases (19/33; 57.8%), and patients with a strong intraoperative fluorescence signal (grade 2) have a GTR in 95 cases (95/142; 66.9%) (Fisher’s exact test (two-sided): p = 0.32). The intake of AEDs has no impact on the overall survival. Patients with a baseline intake of AEDs have a median overall survival of 22.0 months (95% CI = 16.5–27.5), and those without baseline intake of AEDs a median overall survival of 24.0 months (95% CI = 16.5–31.5) (log-rank test: p = 0.79). Patients with the intake of levetiracetam have a median overall survival of 22.0 months (95% CI = 16.6–27.4), whereas those patients who take no AEDs have a median overall survival of 19 months (95% CI = 12.6–25.4; log-rank test: *p* = 0.91)

## 4. Discussion

The use of 5-ALA induced PpIX fluorescence is established in surgery for GB, and since FDA approval in 2017, fluorescence-guided surgery is becoming globally introduced in neurosurgical centers. A 5-ALA-guided surgery was found to significantly enhance the rate of gross total resection compared to conventional white-light microsurgery, and results in a significant improvement in the 6 month progression-free survival rates [16]. Furthermore, there are also emerging data regarding the extension of indications for 5-ALA-guided surgery, such as the application in metastatic tumors and meningiomas [47,48,49]. The objective of the present investigation was to evaluate whether different intraoperative 5-ALA fluorescence quality grades expressed by GB cells are influenced by the intake of antiepileptic drugs. This issue is of paramount importance because the presentation of symptomatic epilepsy as an initial symptom affects between 25% and 50% of all patients with a glioblastoma [29].

Our findings are summarized as follows: the intake of AEDs prior to 5-ALA-guided surgery for IDH1 wild-type GB is a significant and independent risk factor for poor 5-ALA fluorescence quality (fluorescence grade 0 + 1). Our results are predominantly based on patients treated with levetiracetam. Hence, this observed impact of AED treatment on the intraoperative fluorescence quality is reconfirmed in the subgroup of patients treated with either levetiracetam, or no AED. In the present study, we also observe a higher rate of GTR among those patients with a strong 5-ALA fluorescent quality. However, the results do not achieve a statistical significance, and this issue concerning the impact of AEDs on 5-ALA-guided surgical extent of resection has to be investigated in a homogeneous cohort regarding attainability of gross total resection and eloquent location, using an established functional classification system (e.g., Sawaya grading) [50].

Due to the high incidence of symptomatic epilepsy as an initial symptom at presentation, AEDs as medical treatment of seizures are common practice in the healthcare of GB patients [29]. To date, the potential clinical dilemma between the intake of AEDs and intraoperative 5-ALA fluorescence is almost unexplained. The study group of Hefti et al. [27] show in an in vitro investigation that the PpIX synthesis is significantly reduced in glioma cells by the application of phenytoin. Moreover, there are also further data that analyze the anticonvulsants desipramine, phenytoin, valproic acid, or levetiracetam, in combination with or without dexamethasone, and their influence on PpIX production after 5-ALA application in U87MG cells [28]. They found that all those drugs, except levetiracetam, reduce PpIX production in U87MG GB cells. Furthermore, they found that the cellular retention of PpIX is significantly reduced in cells treated with both dexamethasone and phenytoin. In the present study, we observe that dexamethasone is a significant risk factor for poor fluorescence response in the univariable analysis. However, in the multivariable analyses of the entire group, and the subgroup (levetiracetam intake vs. no AED), we observe that only the preoperative intake of AEDs or levetiracetam significantly reduces the intraoperative 5-ALA fluorescence quality. Levetiracetam is the most common prescribed AED in our study cohort, and this finding is predominantly based on the impact of levetiracetam. Hence, levetiracetam might also have a significant effect on the 5-ALA quality in a clinical setting, as reported by the in vitro results for the other medications in the class of anticonvulsants. A recent retrospective study enrolled 27 low-grade glioma patients, and analyzes the effect of AEDs on the visible 5-ALA fluorescence. They also demonstrate that AEDs, including levetiracetam, seem to reduce the presence of visible 5-ALA fluorescence [51]. However, it must be remembered that the majority of pure low-grade gliomas cannot be sufficiently visualized by 5-ALA, and it is predominantly a useful tool to detect solitary areas of anaplastic foci within low-grade gliomas [52,53,54]. A further potential mechanism of levetiracetam disturbing the fluorescence quality and 5-ALA-guided surgery is the alteration of the mitochondrial membrane potential [55]. Therefore, there is a potential disruption of the absorption of exogenously administered 5-ALA into the cytoplasm of the mitochondria in patients who take levetiracetam. Nevertheless, the results of the increase or decrease in the mitochondrial membrane potential by levetiracetam in various cell lines are conflicting. A 5-ALA-guided surgery is a powerful tool, which is known to enhance the extent of resection and improve progression-free survival. However, the signal intensity is also significantly influenced by the IDH1 mutation, as IDH is a known enzyme of the Krebs cycle and catalyzes the formation of α-ketoglutarate from the decarboxylation of D-isocitrate [56]. Hence, our study was focused on IDH1 wild-type GBs, in order to reduce this potential bias. Due to the paramount importance of 5-ALA regarding intraoperative tumor visualization and long-term tumor control, there are also studies which investigate preoperative screening methods to identify patients who will intraoperatively respond to 5-ALA. Utsuki et al. [57] demonstrate urine analysis of PpIX accumulation in GBM patients as a useful tool to determine those patients. However, this model gives no explanation why a subgroup of patients does not accumulate a sufficient amount of PpIX for intraoperative 5-ALA-guided surgery. Levetiracetam was suggested as the first-line AED treatment in glioma patients, due to the low level of toxicity, wide therapeutic index, and lack of hepatic metabolism [58]. Due to the high incidence of symptomatic epilepsy at presentation in glioma patients, further centers have to evaluate our findings and provide an external validation. Moreover, there are several potential intraoperative fluorescence imaging alternatives, such as sodium fluorescein, second window indocyanine green, and 5-aminofluorescein-labeled albumin, which were investigated regarding glioma surgery [59,60,61]. Those fluorescent dyes were demonstrated to aid glioma surgery and improve the extent of resection, compared to conventional white-light surgery. However, there are no prospective high-class data regarding progression-free survival for those dyes compared to the established fluorescent dye 5-ALA. Future comparative trials with 5-ALA have to focus on the fluorescence visualization in patients who are under levetiracetam treatment. Furthermore, the prophylactic use of AED in patients with brain tumors is also highly debated [62]. The introduction of a prophylactic AED treatment is still observed, although the results of meta-analyses and randomized controlled trials do not reveal significantly lower epilepsy incidences than in the control groups [63]. Moreover, the influence of AEDs on overall survival is also increasingly discussed, and the majority of studies do not report a strong impact, despite individual studies showing a benefit of levetiracetam use in GB patients treated with concurrent temozolomide chemoradiotherapy [64,65,66]. Nevertheless, in the present series, we did not find a significant association between intake of AEDs in general or levetiracetam and overall survival. Furthermore, it is unclear whether AEDs potentially influence the overall survival by inhibiting the tumor progression, or by influencing the EoR via the intraoperative fluorescence quality. Hence, physicians have to carefully consider the indication of levetiracetam and the risk–benefit ratio of prophylactic AEDs in patients with a suspected diagnosis of a GB, due to the potential need for 5-ALA-guided surgery in order to perform a maximum cytoreductive surgery, enhancing progression-free survival. Moreover, the use of alternative tools for intraoperative imaging, such as intraoperative MRI or sodium fluorescein in those patients who take levetiracetam, must be considered. 

The present study has several limitations. The main limitation is its retrospective design, and the difficulty to standardize and homogeneously measure fluorescence quality grades. However, we have a highly standardized surgical workflow, with the same operative equipment, and all surgeries were performed in a dark room to reduce the heterogeneity. Furthermore, the present investigation exclusively focused on IDH1 wild-type glioblastomas, due to potential molecular relationships between the IDH genotypes and tumor fluorescence [34]. Future trials have to intraoperatively quantify the protoporphyrin IX levels with an optic fiber spectrometer to validate our findings regarding the influence of levetiracetam [64,67].

## 5. Conclusions

The present investigation analyzed the impact of antiepileptic drugs and levetiracetam on intraoperative 5-ALA fluorescence. The observance of 5-ALA fluorescence is a powerful tool in glioblastoma surgery. According to our series, we observed a significant influence of the intake of levetiracetam in 5-ALA-guided GB surgery, which reduces the 5-ALA fluorescence visualization. The demonstrated issue has to be considered in the clinical setting of glioma patients who frequently have symptomatic epilepsy as initial symptom at presentation. Future trials have to provide an external validation of our findings, and prove this potential bias in alternative fluorescent dyes.

## Figures and Tables

**Figure 1 cancers-14-02134-f001:**
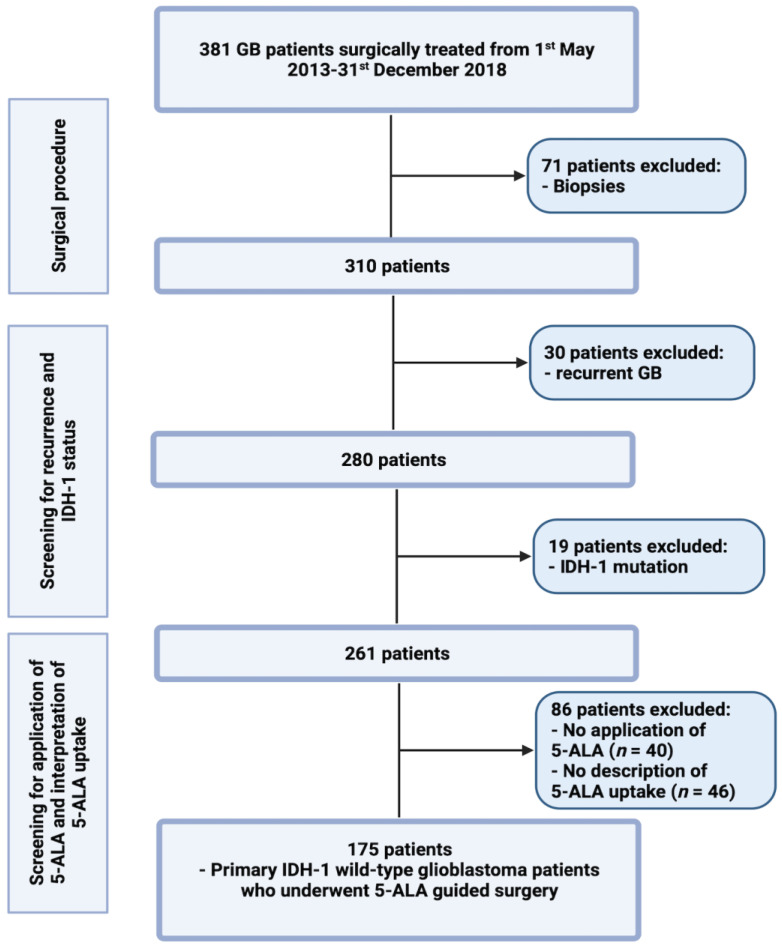
Flow chart illustrating the selection process of consecutive meningioma patients between 1 May 2013 and 1 December 2018.

**Figure 2 cancers-14-02134-f002:**
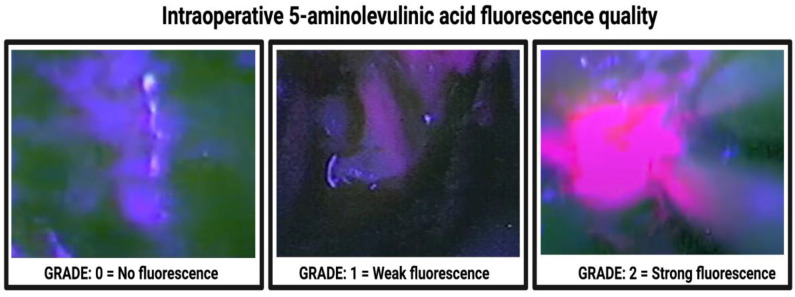
Classification system of 5-ALA fluorescence quality grading.

**Figure 3 cancers-14-02134-f003:**
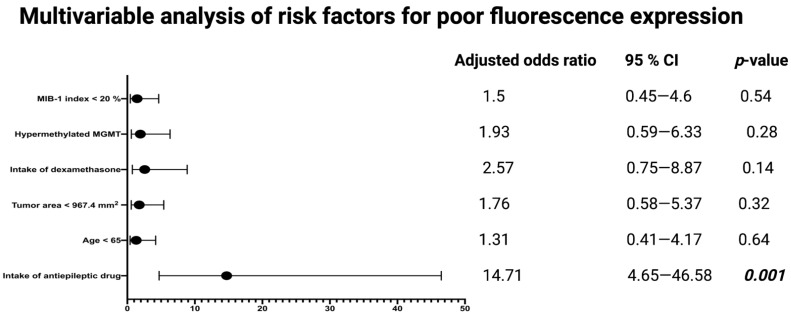
Forest plots from multivariable analysis: intake of antiepileptic drugs is an independent predictor of poor intraoperative 5-ALA fluorescence quality in surgery for IDH1 wild-type glioblastoma. *p*-values in bold and italics display statistically significant results.

**Figure 4 cancers-14-02134-f004:**
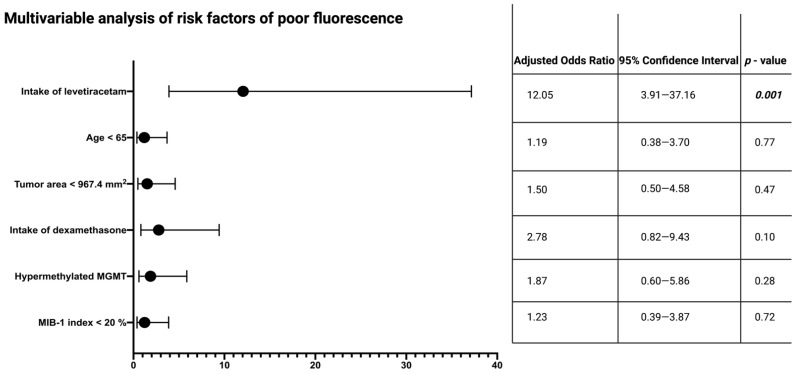
Forest plots from multivariable analysis: intake of levetiracetam is an independent predictor of poor intraoperative 5-ALA fluorescence quality in surgery for IDH1 wild-type glioblastoma. *p*-values in bold and italics display statistically significant results.

**Table 1 cancers-14-02134-t001:** Patient characteristics of IDH1 wild-type glioblastoma (*n* = 175).

Median age (IQR) (in years)	66 (56–73)
Sex	
Female	66 (37.7%)
Male	109 (62.3%)
Median preoperative KPS (IQR)	90 (80–90)
Median body mass index (IQR)	25.7 (23.4–28.7)
Preoperative epilepsy	52 (29.7%)
Generalized	24 (13.7%)
Complex partial	9 (5.1%)
Simple partial	19 (10.9%)
Type of antiepileptic medication	
Levetiracetam	43 (24.6%)
Lamotrigine	2 (1.1%)
Valproate	1 (0.6%)
Carbamazepine	1 (0.6%)
Phenytoin	1 (0.6%)
Benzodiazepines	1 (0.6%)
Dexamethasone intake	
Yes	75 (42.9%)
No	100 (57.1%)
Median tumor area (IQR), mm^2^	1268 (724.8–2113)
Median peritumoral edema (IQR), mm	21.3 (15.3–29.7)
MGMT promoter hypermethylation	69 (39.4%)
Median MIB-1 labeling index (IQR)	15 (10–20)
5-ALA fluorescence grade	
Grade 0—no fluorescence	16 (9.1%)
Grade 1—weak fluorescence	17 (9.7%)
Grade 2—strong fluorescence	142 (81.1%)

**Table 2 cancers-14-02134-t002:** Comparison of patient characteristics among fluorescence quality grades (using Fisher’s exact test (two-sided) and ANOVA).

Characteristics	Fluorescence Grade 0 (*n* = 16)	Fluorescence Grade 1 (*n* = 17)	Fluorescence Grade 2 (*n* = 142)	*p*-Value
Age (years), mean ± SD	56.3 ± 15.9	61.5 ± 11.0	64.1 ± 12.7	0.06
Sex				0.26
Female	3 (18.8%)	6 (35.3%)	57 (40.1%)	
Male	13 (81.2%)	11 (64.7%)	85 (59.9%)	
Preoperative AED				*0.001*
Yes	14 (87.5%)	12 (70.6%)	26 (18.3%)	
No	2 (12.5%)	5 (29.4%)	116 (81.7%)	
Dexamethasone intake				*0.023*
Yes	11 (68.8%)	10 (58.8%)	54 (38.0%)	
No	5 (32.2%)	7 (42.2%)	88 (62.0%)	
Body mass index, mean ± SD	27.3 ± 6.7	26.1 ± 5.5	26.4 ± 3.9	0.72
Tumor area, mean ± SD, mm^2^	834.1 ± 533.8	1197.6 ± 921.4	1553.7 ± 1047.8	*0.03*
Peritumoral edema, mean ± SD, mm	21.1 ± 9.7	21.7 ± 10.6	23.7 ± 11.7	0.64
MGMT promoter status [available in 166 patients]				0.19
Hypermethylated	3 (20.0%)	6 (37.5%)	60 (44.4%)	
Non-hypermethylated	12 (80.0%)	10 (62.5%)	75 (55.6%)	
MIB-1 index, mean ± SD	15.9 ± 6.5	19.7 ± 10.8	17.7 ± 8.2	0.48

AED = Antiepileptic drug; MIB-1 = Molecular immunology borstel-1; SD = Standard deviation. Significant test results are italicized.

**Table 3 cancers-14-02134-t003:** Comparison of patient characteristics among fluorescence quality grades in patients treated with levetiracetam or without AEDs (using Fisher’s exact test (two-sided) and independent *t*-test) (*n* = 169).

Characteristics	Fluorescence Grade 0 (*n* = 12)	Fluorescence Grade 1 (*n* = 16)	Fluorescence Grade 2 (*n* = 141)	*p*-Value
Age (years), mean ± SD	55.0 ± 16.0	61.5 ± 11.4	64.1 ± 12.7	0 vs. 1: 0.22
				1 vs. 2: 0.44
				0 vs. 2: *0.02*
Sex				0.49
Female	3 (25.0%)	5 (31.3%)	57 (40.4%)	
Male	9 (75.0%)	11 (68.8%)	84 (59.6%)	
Preoperative Levetiracetam				*0.001*
Yes	10 (83.3%)	12 (75.0%)	21 (14.9%)	
No	2 (16.7%)	4 (25.0%)	120 (85.1%)	
Dexamethasone intake				*0.001*
Yes	11 (91.7%)	11 (68.8%)	54 (38.3%)	
No	1 (8.3%)	5 (31.1%)	87 (61.7%)	
Body mass index, mean ± SD	27.0 ± 7.5	26.3 ± 5.6	26.4 ± 3.9	0 vs. 1:
				0.80
				1 vs. 2:
				0.95
				0 vs. 2:
				0.81
Tumor area, mean ± SD, mm^2^	830.7 ± 531.8	1274.0 ± 905.5	1550.9 ± 1048.5	0 vs. 1:
				0.18
				0 vs. 2:
				*0.002*
				1 vs. 2:
				0.33
Peritumoral edema, mean ± SD, mm	21.6 ± 11.1	21.3 ± 10.8	23.7 ± 11.7	0 vs. 1:
				0.95
				0 vs. 2:
				0.57
				1 vs. 2:
				0.45
MGMT promoter status [available in 160 patients]				0.18
Hypermethylated	2 (18.2%)	5 (33.3%)	60 (44.8%)	
Non-hypermethylated	9 (81.8%)	10 (66.7%)	74 (55.2%)	
MIB-1 index, mean ± SD	16.6 ± 6.8	19.7 ± 10.8	17.7 ± 8.2	0 vs. 1:
				0.42
				0 vs. 2:
				0.67
				1 vs. 2:
				0.40

AED = Antiepileptic drug; MIB-1 = Molecular immunology borstel-1; SD = Standard deviation. Significant test results are italicized.

## Data Availability

The data presented in this study are available on request from the corresponding author. The data are not publicly available due to privacy and ethical restrictions.

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
