# Peer review of "Impact of Levetiracetam Treatment on 5-Aminolevulinic Acid Fluorescence Expression in IDH1 Wild-Type Glioblastoma"

_cancers, 2022, doi:10.3390/cancers14092134_

Round 1
Reviewer 1 Report
The authors present a very interesting report about the impact of preoperative AEDs on the quality of 5-ALA fluorescence in IDH-1 wild-type GBM surgery. It seems that, at least for wild-type GBM, preoperative AEDs reduce the quality of fluorescence. At our Institution, we have never administered prophylactic anti-seizure therapy, and this could be another argument again such a strategy. However, some authors have recently proposed a role for AEDs in controlling tumor progression ( see e.g. Cucchiara et al., Epileptogenesis and oncogenesis: An antineoplastic role for antiepileptic drugs in brain tumours? Pharmacol Res. 2020 Jun;156:104786. doi: 10.1016/j.phrs.2020.104786). Whereas is not the topic of the paper, it is possible to draw similar considerations in the authors' series?
This is just a minor issue; I think that the manuscript represents a relevant addition to the existing literature about the management of GBM patients
Author Response
Dear Reviewer
Thank you for thoroughly reviewing our manuscript and the comments, which will allow us to improve it to a better scientific level and make it more understandable to the readership.
The reviewer is absolutely right that the potential of AEDs influencing both epileptogenesis and tumorigenesis is increasingly debated. However, we could not identify an impact of AEDs or in particular levetiracetam on the overall survival in our cohort (see section 3.5 Impact of intraoperative fluoresence quality on extent of resection and influence of AEDs, and levetiracetam on overall survival”). Patients with a poor fluorescence quality (grades 0 & 1) had a GTR in 19 cases (19/33; 57.8%), and patients with a strong intraoperative fluorescence signal (grade 2) had a GTR in 95 patients (95/142; 66.9%), respectively (Fisher´s exact test (two-sided): p = 0.32). The intake of AEDs had no impact on the median overall survival. Patients with a preoperative intake of AEDs had a median overall survival of 22.0 months (95% CI = 16.5-27.5), and those without preoperative intake of AEDs a median overall survival of 24.0 months (95% CI = 16.5-31.5), respectively (log-rank test: p = 0.79). Patients with the intake of levetiracetam had a median overal survival of 22 months (95% CI = 16.6-27.4), whereas those patients who took no AEDs had a median overall surivval of 19 months (95% CI = 12.6-25.4; log-rank test: p = 0.91). Furthermore, we have also revised our discussion and included the debate that our findings revealed no significant impact of AEDs on the overall survival. Nevertheless, the reviewer is absolutely right that there are some studies describing a potential impact of AEDs on both epilepsy and tumor progression if they are used in GB patients treated with concurrent temozolomide chemoradiotherapy [1-3].
References
- Knudsen-Baas, K.M.; Engeland, A.; Gilhus, N.E.; Storstein, A.M.; Owe, J.F. Does the choice of antiepileptic drug affect survival in glioblastoma patients? J Neurooncol. 2016, 129(3), 461-469
- Roh, T.H.; Moon, J.H.; Park, H.H.; Kim, E.H.; Hong, C.K.; Kim, S.H.; Kang, S.G.; Chang, J.H..Association between survival and levetiracetam use in glioblastoma patients treated with temozolomide chemoradiotherapy. Sci Rep. 2020,10, 10783
- Cucchiara, F.; Pasqualetti, F.; Giorgi, F.S.; Danesi, R.; Bocci, G. Epileptogenesis and oncogenesis: An antineoplastic role for antiepileptic drugs in brain tumours? Pharmacol Res. 2020, 156, 104786.
Reviewer 2 Report
This clinical study presents some interesting data on antiepileptic drugs effect on 5-ALA guided surgery by fluorescence. The authors present some compelling data but I think the report is unfocussed and does not get to the point well. The authors set up the paper as an investigation of AEDs but in reality most of the data concerns levetiracetam. The other AEDs have only been prescribed to a few patients. The authors should also keep in mind that not all these AEDs function the same. A more general study would have parsed the AEDs and explored their effects independently. However, as the authors are limited by the available patients this was not possible. Thus, I think the authors should reorganize the paper to focus specifically on levetiracetam with some mention of the other drugs. The introduction should be re-written with a focus on levetiracetam. As the authors probably know the mechanism of action of levetiracetam is still under exploration. Their present study would shed some light on the effects of this drug. Additionally, the paper would have a higher impact if it stays focused on this drug. I suggest that the name of the drug should be put in the title of the paper.
Two other changes are suggested below which concern placing the study within known literature:
Line 72: The mechanism of PpIX accumulation in 5-ALA treated patients is not that unclear. In fact, exogeneous 5-ALA is transported into the mitochondria where it activates PpIX expression. The authors should discuss this in the introduction. See for example this paper PMID: 31332208 and references therein.
This sentence at line 74 “For instance, increased metabolism and up-regulation of porphyrin- 74 producing enzymes, reduced metabolism of iron within tumor cells, and reduction of activity of the enzyme ferrochelatase which converts the visible and fluorescing PpIX into heme are highly debated [22-24].” The authors are basing this on old studies. Please review some current literature. The mechanism is not as “highly debated” as the authors claim. There is more work to be done but quite a few things are not debated!! Check out this current review PMID: 35284403
Author Response
Dear Reviewer
Thank you for thoroughly reviewing our manuscript and the comments, which will allow us to improve it to a better scientific level and make it more understandable to the readership.
The reviewer is absolutely right that levetiracetam was the most common prescribed AED. Hence, the key message of this manuscript is predominantly based on the effect of this drug. We have thoroughly revised the manuscript according to the suggestion of the reviewer.
In general, the following points were revised:
- Replacement of AED by levetiracetam in the title of the manuscript and in the graphical abstract
- Revision of the introduction with a focus on levetiracetam
- Creation of a new section “3.4 Specific impact of levetiracetam on intraoperative 5-ALA fluorescence”
- Creation of a new table 3 and figure 4 displaying the univariable and multivariable analysis of risk factors for poor fluorescence in the subgroup of patients who took either levetiracetam or no AEDs.
- Revision of the discussion with a focus on levetiracetam
In the following we explain each point which has been revised:
In the introduction we have revised the review of the mechanism of exogeneous 5-ALA application and added the information regarding the absorption of exogeneous 5-ALA into the cytoplasm of the mitochondria which is further used as a substrate for protoporphyrin IX [1]. AEDs such as levetiracetam were suggested to injure the mitochondrial membranes which potentially results in an inhibition of PpIX synthesis in glioblastoma cells [2, 3]. Symptomatic epilepsy affects between 25% and 50% of all GB patients [4]. The International League Against Epilepsy (ILAE) recommends levetiracetam as a class A efficacy AED [5]. Hence, this potential dilemma between 5-ALA application for GB surgery and AED intake is predominantly caused by levetiracetam. Therefore, the reviewer is absolutely right that the key message is predominantly based on the impact of levetiracetam and this issue is interesting for oncologists, neurologists, radiotherapists, and neurosurgeons who are caring for these vulnerable patients.
In the chapter results we created a new section “3.4 Specific impact of levetiracetam on intraoperative 5-ALA fluorescence” to specify the key message and confirm the results in a subgroup exclusively investigating patients who either took levetiracetam or no AED prior to 5-ALA guided surgery. Furthermore, this section includes the newly created table 3 displaying the comparison of patient characteristics among fluorescence quality grades in patients treated with levetiracetam or no AEDs. Figure 4 displays the multivariable analysis of risk factors in this subgroup analysis (Levetiracetam intake vs. No AEDs). Multivariable binary logistic regression analysis revealed that the intake of levetiracetam (adjusted odds ratio: 12.05, 95% CI: 3.91-37.16, p = 0.001) is an independent risk factor for poor intraoperative 5-ALA fluorescence (fluorescence grades: 0&1) in surgery for GB. Moreover, levetiracetam was investigated regarding the impact of overall survival. Patients who took levetiracetam had a median overall survival of 22 months (95% CI: 16.6-27.4), and those who took no AEDs had a median overall survival of 19 months (95% CI: 12.6-25.4), respectively (log-rank test: p = 0.91). The section discussion was revised to clarify that our results are predominantly based on the impact of levetiracetam. A further potential mechanism of levetiracetam influencing the exogeneous application of 5-ALA might be the alteration of the mitochondrial membrane potential by levetiracetam [6]. This might disturb the PpIX synthesis by disturbing the absorption into the cytoplasm of the mitochondria. However, further investigations have to clarify the exact mechanism of levetiracetam influencing the metabolism of 5-ALA in IDH-1 wild-type GB. Furthermore, future comparative studies of fluorescent dyes (sodium fluorescein, 5-ALA) might have a special focus on the subgroup of patients who take levetiracetam prior to surgery. If further trials provide an external validation of our findings, it might has to be considered to additionally use alternative tools for intraoperative imaging such as intraoperative MRI or sodium fluorescein in those patients who take levetiracetam.
References
- Shimura, M.; Nozawa, N.; Ogawa-Tominaga, M.; Fushimi, T.; Tajika, M.; Ichimoto, K.; Matsunaga, A.; Tsuruoka, T.; Kishita, Y.; Ishii, T.; Takahashi, K.; Tanaka, T.; Nakajima, M.; Okazaki, Y.; Ohtake, A.; Murayama, K. Effects of 5-aminolevulinic acid and sodium ferrous citrate on fibroblasts from individuals with mitochondrial diseases. Sci Rep. 2019, 9(1), 10549.
- Hefti, M.; Albert, I.; Luginbuehl, V. Phenytoin reduces 5-aminolevulinic acid-induced protoporphyrin IX accumulation in malignant glioma cells. J Neurooncol. 2012, 108, 443-450
- Lawrence, J.E.; Steele, C.J.; Rovin, R.A.; Belton, R.J.; Belton, R.J. Jr.; Winn, R.J. Dexamethasone alone and in combination with desipramine, phenytoin, valproic acid or levetiracetam interferes with 5-ALA-mediated PpIX production and cellular retention in glioblastoma cells. J Neurooncol. 2016, 127(1), 15-21
- Toldeo, M.; Sarria-Estrada, S.; Quintana, M.; Maldonado, X.; Martinez-Ricarte, F.; Rodon, J.; Auger, C.; Salas-Puig, J.; Santamraina, E.; Martinez-Saez, E. Prognostic implications of epilepsy in glioblastomas. Clin Neurol Neurosurger. 2015, 139, 166-171
- Glauser, T.; Ben-Menachem, E.; Bourgeois, B.; Cnaan, A.; Guerreiro, C.; Kälviäinen, R.; Mattson, R.; French, J.A.; Perucca, E.; Tomson, T.; ILAE Subcommission on AED Guidelines. Updated ILAE evidence review of antiepileptic drug efficacy and effectiveness as initial monotherapy for epileptic seizures and syndromes. Epilepsia. 2013, 54(3), 551-63
- Rogers, S.K.; Shapiro, L.A.; Tobin, R.P.; Tow, B.; Zuzek, A.; Mukherjee, S.; Newell-Rogers, M.K. Levetiracetam Differentially Alters CD95 Expression of Neuronal Cells and the Mitochondrial Membrane Potential of Immune and Neuronal Cells in vitro. Front Neurol. 2014, 18, 5:17
Round 2
Reviewer 2 Report
The authors have made significant changes to this version. This reviewer is satisfied.